# Depletion of Gut Microbiota Inhibits Hepatic Lipid Accumulation in High-Fat Diet-Fed Mice

**DOI:** 10.3390/ijms23169350

**Published:** 2022-08-19

**Authors:** Hui Han, Mengyu Wang, Ruqing Zhong, Bao Yi, Martine Schroyen, Hongfu Zhang

**Affiliations:** 1State Key Laboratory of Animal Nutrition, Institute of Animal Science, Chinese Academy of Agricultural Sciences, Beijing 100193, China; 2Precision Livestock and Nutrition Unit, Gembloux Agro-Bio Tech, University of Liège, 4000 Gembloux, Belgium

**Keywords:** gut microbiota, liver, lipid metabolism, antibiotics cocktail (Abx), high-fat diet (HFD)

## Abstract

Dysregulated lipid metabolism is a key pathology in metabolic diseases and the liver is a critical organ for lipid metabolism. The gut microbiota has been shown to regulate hepatic lipid metabolism in the host. However, the underlying mechanism by which the gut microbiota influences hepatic lipid metabolism has not been elucidated. Here, a gut microbiota depletion mouse model was constructed with an antibiotics cocktail (Abx) to study the mechanism through which intestinal microbiota regulates hepatic lipid metabolism in high-fat diet (HFD)-fed mice. Our results showed that the Abx treatment effectively eradicated the gut microbiota in these mice. Microbiota depletion reduced the body weight and fat deposition both in white adipose tissue and liver. In addition, microbiota depletion reduced serum levels of glucose, total cholesterol (TC), low-density lipoproteins (LDL), insulin, and leptin in HFD-fed mice. Importantly, the depletion of gut microbiota in HFD-fed mice inhibited excessive hepatic lipid accumulation. Mechanistically, RNA-seq results revealed that gut microbiota depletion changed the expression of hepatic genes involved in cholesterol and fatty acid metabolism, such as *Cd36*, *Mogat1*, *Cyp39a1*, *Abcc3*, and *Gpat3*. Moreover, gut microbiota depletion reduced the abundance of bacteria associated with abnormal metabolism and inflammation, including *Lachnospiraceae*, *Coriobacteriaceae_UCG-002*, *Enterorhabdus*, *Faecalibaculum*, and *Desulfovibrio*. Correlation analysis showed that there was strong association between the altered gut microbiota abundance and the serum cholesterol level. This study indicates that gut microbiota ameliorates HFD-induced hepatic lipid metabolic dysfunction, which might be associated with genes participating in cholesterol and fatty acid metabolism in the liver.

## 1. Introduction

The liver plays a central role in lipid metabolism. Dysfunction of hepatic lipid metabolism can cause metabolic diseases, such as non-alcoholic fatty liver disease (NAFLD), obesity, and type 2 diabetes, which have become the most common and costly chronic disorders globally [1].

A large body of evidence has revealed the critical role of gut microbiota in maintaining host lipid metabolism [2,3,4]. For example, patients with NAFLD had lower microbial diversity than healthy controls [5,6]. Additionally, alterations in the gut microbiota community have been identified to accelerate the development of dysbiosis of hepatic lipid metabolism in both clinical and animal studies [7,8,9,10]. Several studies reported that the Firmicutes/Bacteroidetes ratio was elevated in mice fed a high-fat diet (HFD) that induced obesity and NAFLD [11,12], while in another study, a reduced Firmicutes/Bacteroidetes ratio was reported [13]. As for the certain bacteria at the genus level, NAFLD is frequently associated with increased abundance of *Escherichia*, *Dorea*, and *Peptoniphilus* and decreased *Prevotella*, *Faecalibacterium*, *Eubacterium*, and *Coprococcus* [14,15]. However, another study showed that children with obesity and NAFLD exhibited increased *Prevotella* compared to those without NAFLD [16]. The obvious opposite results indicate that the composition of intestinal microbiota is dynamic and can be shaped in different physical conditions.

Furthermore, microbial metabolites, such as short-chain fatty acids (SCFAs), bile acids (BAs), amino-acid-derived microbial metabolites, and even liposaccharide (LPS, the major component of the cell membrane of Gram-negative bacteria) play an important role in modulating lipid metabolism through regulating pathways related to lipid absorption, synthesis, and clearance [4,17,18]. For example, SCFAs, the main metabolites produced by microbial fermentation of dietary carbohydrate, can act as signaling molecules to regulate the expression of genes related to lipogenesis, fatty acid oxidation, and adipogenesis [18]. What is more, it has been found that SCFAs can accelerate the progress of obesity via elevating intestinal energy harvesting, but also reduce energy intake via inhibiting appetite [19]. Therefore, microbiota-derived components or metabolites are complex and precisely modulated in response to genes/pathways in lipid metabolism and the relationship between gut microbiota and host hepatic lipid metabolism needs further investigations.

A recent study showed that treatment with the combination of vancomycin, metronidazole, neomycin, and ampicillin through drinking water can effectively eradicate gut microbiota in mice [20]. Moreover, the combination of these four antibiotics was widely used to investigate the effects of targeted gut microbiota on host metabolism [21,22]. Mice with gut microbiota depletion via an antibiotics cocktail (Abx) treatment still have gut microbiota residues to sustain development of the immune system and metabolism, which are different from germ-free mice [23]. Thus, mice with gut microbiota depletion treated with Abx can be used as a proper model to explore the specific role of gut microbiota on host metabolism. With this model, a study by Zarrinpar et al. (2018) showed that depleted microbiota potentially regulated glucose homeostasis by promoting the utilization of glucose by the colonocytes and by elevating hepatic glucose production and transport in mice [23]. Another study showed that Abx-induced microbiota depletion alleviated insulin resistance and inhibited hepatic lipid accumulation and lipogenesis through modulating bile acid metabolism in HFD-fed hamsters [24].

The gut microbiota and its metabolites involved in hepatic metabolism and inflammation via the gut–liver axis were summarized in our previous review [4]. In particular, gut microbial metabolites, such as SCFAs and BAs can directly affect hepatic glucose and lipid metabolism [25,26,27]. However, the effects of gut microbiota depletion via Abx treatment on hepatic lipid metabolism in mice are not entirely clear yet. Thus, in the current study, the effects of the gut microbiota on hepatic lipid metabolism using an Abx-induced pseudo-germ-free mouse model were investigated.

## 2. Results

### 2.1. Gut Microbiota Composition Was Altered by Abx Treatment in HFD-Fed Mice

To determine the effect of gut microbiota on host lipid metabolism, gut microbiota was depleted by Abx treatment in HFD-fed mice (Figure 1A). 16S rRNA sequencing analysis of mice feces was performed to investigate the altered microbiota community in the intestine. After removing the low-quality sequences, in total, 1,282,709 clean tags were clustered into operational taxonomic units (OTUs) based on 97% identity. The Venn diagram showed that the Chow, HFD, and HFD+Abx groups contained 347, 450, and 211 OTUs, respectively (Figure 1B). Additionally, there were 274 common OTUs between the chow and HFD groups and 100 common OTUs between the HFD+Abx group (Figure 1B), suggesting that there was the biggest similarity between the Chow and HFD. The α-diversity results showed that the HFD had no significant effects on the Sobs, Shannon, Ace, Chao1, and Simpson indexes compared to the chow diet (*p* > 0.05) (Figure 1C–G). However, Sobs, Shannon, Ace, and Chao1 indexes were significantly lower and the Simpson index was significantly higher in the HFD+Abx group compared with the HFD group (*p* < 0.05) (Figure 1C–G). These data suggested that Abx treatment reduced the microbial richness and diversity. Moreover, β-diversity analysis was conducted using principal component analysis (PCA), principal coordinates analysis (PCoA), and non-metric multidimensional scaling (NMDS) methods. The results showed that although the microbiota cluster in mice of the Chow and the HFD group was relatively similar, the microbial community structure was significantly different between the HFD and HFD+Abx groups (Figure 1H–J).

Specifically, the relative abundance results showed that at the phylum level, Firmicutes was the most abundant, followed by Bacteroidetes and Actinobacteriota in the Chow group and Actinobacteriota in the HFD group (Figure 2A). These phyla accounted for the majority of the gut microbiota abundances in the Chow (96.19%) and HFD (94.91%) groups (Figure 2A). However, in the mice of the HFD+Abx group, Proteobacteria (78.23%) was the most abundant, followed by Firmicutes (20.70%) (Figure 2A). In addition, the relative abundances of Bacteroidetes and Verrucomicrobiota were significantly lower in the mice fed an HFD than those in mice fed a chow diet (*p* < 0.05) (Figure 2A). Abx treatment induced a significant decrease in the relative abundance of Firmicutes, Actinobacteria, and Desulfobacterota, while it increased the relative abundance of Proteobacteria in HFD-fed mice (*p* < 0.05) (Figure 2A). In addition, at the genus level, mice fed an HFD had significantly lower relative abundance of *nonrank_f_Muribaculaceae*, *Alistipes*, *Odoribacter*, *Parabacteroides*, *Rikenellaceae_RC9_gut_group*, *Turicibacter*, *Marvinbryantia*, *Monoglobus*, *Euacterium_nodatum_group*, and *Akkermansia* than those fed a chow diet (*p* < 0.05) (Figure 2B,C). Meanwhile, HFD markedly increased the relative abundance of *Roseburia*, which was restored by Abx treatment (*p* < 0.05) (Figure 2B,C). In HFD-fed mice, Abx treatment significantly decreased the relative abundance of *Coriobacteriaceae_UCG-002*, *Enterorhabdus*, *Desulfovibrio*, *Faecalibaculum*, *Lachnoclostridium*, *nonrank_f_Lachnospiraceae*, *noran_f_Ruminococcaceae*, *Eubacterium_brachy_group*, *norank_f_norank_o_Clostridia_UCG-014*, while it increased the relative abundance of *Enterobacter* (*p* < 0.05) (Figure 2B,C).

### 2.2. LPS and SCFA Concentrations

LPS and SCFAs are microbial synthesized components and metabolites and were investigated to confirm the altered composition of intestinal microbiota. The results showed that the serum concentration of LPS was significantly higher in the mice fed an HFD than those fed a chow diet, which was significantly restored by gut microbiota depletion (*p* < 0.05) (Figure 3A). In addition, as shown in Figure 3B–F, HFD failed to affect the concentration of fecal SCFAs compared with the normal chow diet (*p* > 0.05). Gut microbiota depletion induced no detectable butyrate and isovalerate in HFD-fed mice (Figure 3D,F). However, the levels of fecal acetate, propionate, and iso-butyrate were not significantly different between the HFD-fed mice with and without gut microbiota depletion (*p* > 0.05) (Figure 3B,C,E).

### 2.3. Gut Microbiota Depletion Reduced Body Weight and Energy Intake in Mice with HFD

After 3 weeks of feeding, mice fed an HFD showed significantly higher body weight than those fed a chow diet (*p* < 0.05) (Figure 4A,B). From the first week to the end of the experiment, Abx treatment significantly decreased the body weight in HFD-fed mice (*p* < 0.05) (Figure 4B). In addition, mice fed an HFD had markedly higher energy intake than those fed a chow diet during the whole experiment (*p* < 0.05) (Figure 4C). Abx treatment significantly decreased the energy intake from the first to the third week in HFD-fed mice (*p* < 0.05) (Figure 4C). 

### 2.4. Gut Microbiota Depletion Alleviated Serum Lipid Metabolic Parameters in HFD-Fed Mice

Serum total cholesterol (TC), triglycerides (TG), low-density lipoprotein cholesterol (LDL), high-density lipoprotein cholesterol (HDL), and free fatty acid (FFA) levels were examined as representative lipid metabolic parameters. Compared with the Chow group, mice in the HFD group had significantly higher serum HDL levels (*p* < 0.05), while there was no significant difference in serum TC, TG, LDL, and FFA levels between the Chow and HFD groups (*p* > 0.05) (Figure 5A–E). However, microbial depletion significantly decreased the serum TC and LDL levels in HFD-fed mice (*p* < 0.05) (Figure 5B,D).

In addition, HFD significantly lowered the blood glucose concentration (*p* < 0.05) (Figure 5F), while it had no significant effects on the fasting blood glucose level (*p* > 0.05) (Figure 5G) compared with chow diet. However, ablation of gut microbiota markedly reduced the blood glucose and fasting blood glucose concentrations in HFD-fed mice (*p* < 0.05) (Figure 5F,G), suggesting that gut microbiota depletion modulated glucose metabolism. We then investigated the serum concentrations of leptin and insulin. The results showed that HFD significantly increased the serum leptin and insulin levels, which were alleviated by the depletion of gut microbiota (*p* < 0.05) (Figure 5H,I). 

### 2.5. Gut Microbiota Depletion Reduced Fat Deposition in Obese Mice

To determine the effect of gut microbiota depletion on body fat deposition, inguinal white adipose tissue (iWAT), representing the subcutaneous adipose tissue, and perirenal adipose tissue (pWAT) and epididymal white adipose tissue (eWAT), which represent the visceral adipose tissues, were investigated. The results showed that the ratios of the mass of iWAT, pWAT, and eWAT, to total body weight in mice fed an HFD were significantly higher than those fed a chow diet, which were markedly restored by gut microbiota depletion (*p* < 0.05) (Figure 6A,B). We hypothesize that the weight loss induced by microbiota depletion mainly comes from a reduced WAT mass.

Besides adipose tissue, liver, the central organ for lipid metabolism, is sensitive to HFD, and rapidly develops lipid accumulation [28]. Our present study showed that HFD induced a significant increase in the liver weight (*p* < 0.05), while failed to affect the relative weight of liver (*p* > 0.05) (Figure 6C,D). Gut microbiota ablation mediated by Abx treatment tended to decrease the weight and relative weight of liver in HFD-fed mice, although the difference was not significant (*p* > 0.05) (Figure 6C,D). Results from H&E staining indicated that, compared with mice fed a chow diet, mice fed an HFD displayed an excessive hepatic lipid accumulation, which is characterized by more lipid droplets (Figure 6E). In contrast, microbiota depletion inhibited hepatic lipid accumulation in HFD-fed mice (Figure 6E). These data demonstrated that gut microbial depletion reduced hepatic lipid accumulation in HFD-fed mice.

### 2.6. Gut Microbiota Depletion Modulated the Expression of Genes Involved in Lipid Metabolism in the Liver

The protective effect of gut microbial depletion against HFD-induced excessive hepatic lipid accumulation was further investigated in a more mechanistic manner. The gene expression profile in the liver detected by RNA sequencing is shown in Figure 7. The PCA result showed a unique clustering of Chow, HFD, and HFD+Abx samples (Figure 7A). A total of 154 genes were significantly differentially expressed, with 35 being upregulated and 119 being downregulated in mice fed an HFD compared to those fed a chow diet (Figure 7B,D). Meanwhile, compared with the HFD treatment, a total of 76 genes were significantly differentially expressed, with 21 being upregulated and 55 being downregulated in mice treated with HFD+Abx (Figure 7C,D). As shown in Figure 7E,F, compared with chow diet, HFD mainly altered the expression of genes related to lipid metabolism, such as *mevalonate diphosphate decarboxylase* (Mvd), *3-hydroxy-3-methylglutaryl-CoA synthase 1* (Hmgcs1), *proprotein convertase subtilisin/Kexin type 9* (Pcsk9), *stearoyl-CoA desaturase* (Scd1), and *ATP-binding cassette subfamily B member 1* (Abcb1). Similarly, gut microbiota depletion also mainly changed the expression of genes associated with lipid metabolism, such as *cytochrome P450 family 39 subfamilies A polypeptide 1* (Cyp39a1), *ATP-binding cassette subfamily C member 3* (Abcc3), *fatty acid translocase* (Fat, also known as Cd36), and *Acyl-CoA:glycerol-3-phosphate acyltransferase* (Gpat3) (Figure 7E,F). Notably, HFD significantly upregulated the expression of *Cyp39a1*, which was restored by gut microbiota depletion (Figure 7E,F).

The differentially expressed genes (DEGs) between the Chow, HFD, and HFD+Abx groups were analyzed based on Gene Ontology (GO) (Figure 8A,B). The results of DEGs between the Chow and HFD groups showed enriched GO terms in the three categories, Biological Process (BP), Cellular Component (CC), and Molecular Function (MF). Cellular process, biological regulation, and metabolic process were the most enriched in the BP category; cell part, organelle, and membrane were the most enriched for the CC category; binding, catalytic activity, and transporter activity were the top 3 items in the MF category (Figure 8A). In addition, the results of DEGs between the HFD-fed mice with and without microbiota depletion showed that in the BP category the top 3 items remained cellular process, biological regulation, and metabolic process. Additionally, for the CC category the same top terms were enriched, namely cell part, organelle, and membrane, as well as for the MF category, with binding, catalytic activity, and transporter activity (Figure 8B).

The results of the KEGG pathway annotation analysis revealed that, compared to the chow diet, genes upregulated by HFD were mainly involved in lipid metabolism, metabolism of terpenoids and polyketides, and endocrine system (Figure 8C). Genes downregulated by HFD were mainly involved in the endocrine system, signal transduction, and lipid metabolism (Figure 8C). Gut microbiota depletion mainly increased the genes associated with lipid metabolism, metabolism of cofactors and vitamins, nervous system, sensory system, and cancer in HFD-fed mice (Figure 8D). In addition, microbiota depletion mainly decreased the genes involved in lipid metabolism, signal transduction, endocrine system, carbohydrate metabolism, and metabolism of cofactor and vitamins in HFD-fed mice (Figure 8D).

### 2.7. Validation of Gene Expression by qRT-PCR

To confirm the RNA-seq data, DEGs involved in lipid metabolism, including *Scd1*, *fatty acid synthase* (Fasn), *Gpat3*, *Abcb1b*, *Abcc3*, *Pcsk9*, *Mvd*, *Hmgcs1*, and *Cyp39a1*, were further validated by qRT-PCR in the three groups of mouse livers (Figure 9). Consistent with the RNA-seq results, qPCR results showed that compared to chow diet, HFD downregulated the mRNA expression levels of *Scd1* and *Cyp39a1* but upregulated *Fasn* (*p* < 0.05) (Figure 9). Meanwhile, microbial depletion decreased the mRNA expression levels of *Gpat3* and *Abcc3*, while increased the mRNA expression levels of *Psk9*, *Mvd*, and *Hmgcs1* in HFD-fed mice (*p* < 0.05) (Figure 9).

### 2.8. Correlation Analysis between Gut Microbiota and Serum Lipid Metabolic Parameters

A Spearman correlation between gut microbiota and serum lipid metabolic parameters showed that the abundance of *Turicibacter* was negatively correlated with body weight (*p* < 0.05) (Figure 10). The abundance of *Enterobacter* was significantly negatively correlated with the serum levels of TC and LDL, while the abundance of *unclassified_f_Lachnospiraceae*, *Coriobacteriaceae_UCG-002*, *Anaerotruncus*, *Lachnoclostridium*, *Desulfovibrio*, *norank_f_Lachnospiraceae*, *Blautia*, *Lachnospiraceae_NK4A136_group*, *norank_f_Ruminococcaceae*, *Roseburia*, and *Lachnospiraceae_UCG-006* were significantly positively correlated with the serum levels of TC and LDL (*p* < 0.05) (Figure 10). In addition, there were significantly positive associations between the abundance of *Faecalibaculum*, *Enterorhabdus*, and *nonrank_f_Erysipelotrichaceae* and serum TG level (*p* < 0.05) (Figure 10). The abundances of *Blautia* and *Roseburia* were significantly positively correlated with serum leptin level (*p* < 0.05). Meanwhile, the abundances of *Lactobacillus* and *Enterococcus* were significantly positively correlated with serum insulin level (*p* < 0.05) (Figure 10). These data demonstrated there were strong associations between the gut microbiota and serum lipid metabolic parameters, especially the serum TC and LDL.

## 3. Discussion

Metabolic diseases, such as obesity and NAFLD, are regarded as the most serious public health problems worldwide [29]. Mounting evidence has shown a strong association between gut microbiota and hepatic metabolic disorders, including obesity, diabetes, and NAFLD [3,4,30,31]. Therefore, strategies to alter the gut microbiome have been used for the treatment of metabolic disorders in clinical studies [32,33,34]. It is vital to understand the mechanism of gut microbiota in regulating host hepatic metabolism, especially lipid metabolism. Mice intestinal microbiota depletion established by Abx treatment, also called pseudo-germ-free mice, has been widely used for exploring the role of gut microbiota in the host. In this study, we investigated the effects of gut microbiota on the host hepatic lipid metabolism by depleting gut microbiota via an Abx treatment in HFD-fed mice.

The present data showed that HFD had no significant effect on the richness and diversity of gut microbiota when compared to chow diet, which is consistent with a previous study [35]. On the other hand, an Abx treatment reduced gut microbial diversity and richness shown by OTUs and α-diversity analyses, indicating that an Abx treatment could effectively abolish gut microbiota. Previous studies have demonstrated that Abx-depleted microbiota had beneficial metabolic effects on the host, which is characterized by a decreased body weight and blood glucose level in HFD-fed mice [36,37]. Consistently, our present study showed that an HFD induces a higher body weight, WAT weight, and blood glucose levels, which were alleviated by microbiota depletion. Furthermore, HFD commonly induced hyperlipidemia, which is a serious cause of metabolic disorders. In the present study, we found that the absence of gut microbiota reduced the serum levels of TC and LDL in HFD-fed mice. These data demonstrate that gut microbiota depletion was involved in alleviating HFD-induced obesity.

It is well established that insulin plays a crucial role in modulating glucose and lipid metabolism by involving hepatic glucose uptake and lipid synthesis and storage [38]. Leptin is synthesized primarily by adipose tissue and the circulating leptin levels are typically proportional to body fat mass [39]. Leptin can modulate glucose and lipid metabolism and inhibit feed intake by directly acting on the central nervous system, which collectively regulates the metabolism of the host [40]. HFD usually induces elevated serum insulin and leptin levels [41,42,43], which are the indicators of insulin and leptin resistance, respectively, and risk factors for metabolic disorders [44]. Partial leptin reduction improved leptin and insulin sensitivity and decreased body weight in mice with HFD-induced obesity [44]. Our present results showed that the absence of gut microbiota ameliorated HFD-elevated serum insulin and leptin levels in mice, which further confirmed that microbial abolishment has beneficial metabolic effects in obese mice fed an HFD.

The liver plays a central role in controlling systemic glucose and lipid homeostasis by modulating pathways in glucose and lipid metabolism, such as glycogenesis, glycolysis, lipogenesis, cholesterol metabolism, and fatty acid oxidation [45,46]. In the present study, microbiota depletion reduced hepatic lipid accumulation in HFD-fed mice. Liver RNA-seq analysis showed that HFD upregulated the expression levels of *Hmgcs1*, *Mvd*, and *Idi1*, also confirmed by qPCR, which have been reported to be involved in de novo cholesterol biosynthesis [47]. In addition, HFD downregulated the expression of *Abcg4*, which can promote the efflux of cellular cholesterol to HDL and inhibit hepatic cholesterol accumulation [48]. These data demonstrated that HFD caused damage to cholesterol metabolism, which might contribute to hepatic lipid accumulation. Meanwhile, in this study, gut microbiota depletion upregulated the expression of genes in the cytochrome P450 (CYP) family, including *Cyp2c40*, *Cyp2c70*, *Cyp2c68*, and *Cyp39a1*. Cyp2c40 has been reported to be involved in the metabolism of linolenic acid generating epoxy-eicosatrienoic acids [49], which has an anti-inflammatory capacity and exhibits protective effects on metabolic diseases via promoting fatty acid oxidation [50,51,52,53]. Cyp2c70 can metabolize chenodeoxycholic acid into rodent-specific hydrophilic muricholic acid, which can inhibit intestinal cholesterol absorption by decreasing the solubilization of intraluminal micellar cholesterol in mice [54,55]. Cyp39a1 is involved in the alternative bile aid synthesis [56], which plays a critical role in removing cholesterol and inhibiting the sterol accumulation in the liver [57]. Moreover, clinical studies showed that patients with hepatocellular carcinomas (HCC) had lower liver Cyp39a1 expression [58,59]. We found that HFD downregulated the expression of *Cyp39a1*; however, the ablation of gut microbiota upregulated the expression of *Cyp39a1*. Bile acid homeostasis is not only due to the bile acid biosynthesis, but bile acid transport as well. It has been reported that double knockout of *Abcb1a* and *Abcb1b* decreased circulating bile acid concentration in mice [60]. In addition, multidrug-resistance-associated proteins 3 (MRP3/Abcc3) is considered as a bile salt efflux pump and involved in the excretion of bile acids to the circulation [61,62]. In this study, the results showed that HFD downregulated the expression of *Abcb1b*, while microbial depletion downregulated the expression of *Abcc3*. These data suggest that gut microbiota depletion might contribute to cholesterol homeostasis.

In addition, gut microbiota depletion downregulated the expression of *Mogat1* and *Gpat3*, which contribute to the synthesis of TG [63,64,65]. It has been reported that HFD-induced and genetically obese mice had higher expression of *Mogat1* [66]. Moreover, overexpression of *Mogat1* promoted the TG synthesis, while knockdown of liver *Mogat1* decreased body weight, improved insulin sensitivity, and alleviated hepatic lipid accumulation and steatosis in HFD-induced and genetic obese mice [67]. Furthermore, a previous study showed that PCSK9 promoted the degradation of CD36 and then decreased hepatic TG content and fatty acid uptake in mice [68]. In this regard, we found that gut microbiota depletion upregulated the expression level of *Psck9*, while downregulated the expression of *Cd36*. CD36 is known as a fatty acid translocase associated with enhancing hepatic fatty acid uptake and then causes toxicity, inflammation, and apoptosis in hepatocytes; therefore, driving the development of NAFLD [69,70]. These data suggest that microbiota depletion might alleviate hepatic lipid accumulation via inhibiting CD36-mediated hepatic fatty acid uptake in HFD-fed mice. The decreased fatty acid uptake is accompanied by altered expressions of genes associated with fatty acid β-oxidation, including downregulated *Acaa1b* and *Ehhadh* after microbiota depletion via Abx treatment. Taken together, the RNA-seq data suggest that gut microbiota depletion might play a protective role in ameliorating the development of the fatty liver by modulating cholesterol and fatty acid metabolism.

As for microorganism species involved in hepatic cholesterol and fatty acid metabolism, using 16S rRNA amplicon sequencing, we found that at the genus level, a detailed analysis showed that HFD dramatically decreased the abundance of *Alistipes*, *Odoribacter*, *Parabacteroides*, *norank_f_Muribaculaceae*, and *Rikenellaceae_RC9_gut_group*, which all have been reported to have the ability of producing SCFAs [71,72,73,74,75,76]. In addition, our present study showed that HFD induced lower abundance of *Turicibacter* and *Akkermansia*, which have anti-inflammatory and beneficial metabolic effects [77,78,79,80,81]. However, we found that HFD decreased abundance of *Rikenellaceae*, which was contrary to other studies [82,83]. *Roseburia* is a butyrate-producing bacteria and has been considered to have anti-obesity effects [84]. A previous study showed that HFD decreased the abundance of *Roseburia*, which was restored by antibiotic pretreatment in male mice [37]. However, Abx pretreatment decreased the abundance of *Roseburia* in female HFD-fed mice fed [37]. Our study found an increased abundance of *Roseburia* in the HFD group. However, Abx treatment decreased the abundance of *Roseburia,* which might be an explanation for the undetectable butyrate levels in the HFD+Abx group. These inconsistent data might be due to the different sex and age of mice and Abx treatment protocol between the present and previous studies, which needs further investigations. Meanwhile, in this study, we found that Abx treatment decreased the abundance of the genus (*Lachnoclostridium* and *nonrank_f_Lachnospiraceae*) belonging to the *Lachnospiraceae* family. The *Lachnospiraceae* has been reported to be a dominant family in HFD-fed mice [85], suggesting that *Lachnospiraceae* might contribute to abnormal metabolism. *Coriobacteriaceae_UCG-002* and *Enterorhabdus* are members of the *Coribacteriaceae* family, which has been reported to be involved in promoting intestinal cholesterol absorption and positively associated with hepatic TG levels [86,87,88]. Here, we observed that Abx treatment decreased the abundance of *Coriobacteriaceae_UCG-002* and *Enterorhabdus*, which might contribute to lower serum TC levels by inhibiting cholesterol uptake. Moreover, the abundance of *Coriobacteriaceae_UCG-002* and *Enterorhabdus* were significantly positively correlated with serum TC and TG levels, respectively. In addition, Abx treatment also decreased the abundance of *Faecalibaculum* and *Desulfovibrio*, which contribute to the production of pro-inflammatory cytokines [89,90,91,92] and then promote the development of metabolic dysfunction. Similarly, it has been demonstrated that *Desulfovibrio* can produce LPS and then induce inflammation and insulin resistance [91,92]. A recent study showed that an HFD elevated the intestinal and fecal levels of LPS, which might be due to the increased LPS-producing bacteria and damaged intestinal integrity induced by HFD [93,94]. Similarly, our present study showed that an HFD increased the serum LPS levels. Additionally, Abx treatment decreased the serum LPS levels in HFD-fed mice, which might be because of the downregulated abundance of *Desulfovibrio*. Notably, Abx treatment markedly increased the abundance of *Enterobacter*, which might be because it is a major antibiotic-resistant genus [95,96]. Taken together, these data suggest that Abx treatment mainly induced the alterations in gut microbiota involved in hepatic fatty acid and cholesterol metabolism and inflammatory responses in HFD-fed mice, which were also confirmed by differential expression analysis using hepatic RNA sequencing.

## 4. Materials and Methods

### 4.1. Reagents, Mice, and Ethics

The antibiotics were purchased from Meilun Bio (Dalian, China). The standard chow diet (Chow, 10% of energy from fat, D12450B) and high-fat diet (HFD, 60% of energy from fat, D12492) were purchased from Beijing Keao Xieli Feed Co., Ltd. (Beijing, China) and the composition of the diets is shown in Table 1. Six-week-old male C57BL/6J mice were purchased from the Peking University Health Science Center (Beijing, China). Mice (two mice per cage) were housed in standardized environment conditions (room temperature, 22 ± 2 °C; 12/12 h light/dark cycle), with free access to control food and water. All procedures used in this experiment were approved by the Experimental Animal Welfare and Ethical Committee of the Institute of Animal Science, Chinese Academy of Agricultural Sciences.

### 4.2. Mouse Experiments and Sampling

After acclimatization for 1 week, the mice were randomly divided into 3 groups (n = 8 per group) and labeled as Chow, HFD, and HFD+Abx, respectively. Groups HFD and HFD+Abx were fed an HFD, whereas group Chow was fed a standard chow diet for 5 weeks. Mice in HFD+Abx groups received drinking water supplemented with an antibiotic cocktail (Abx: 1 g/L ampicillin, 1 g/L neomycin, 1 g/L metronidazole, and 0.5 g/L vancomycin hydrochloride) during the whole experimental period, according to previous studies [20,23]. Mice in the Chow and HFD groups were given normal drinking water without antibiotics. Water bottles were changed twice weekly to contain fresh Abx (experimental design is shown in Figure 1A). Body weight and feed intake were recorded weekly. At the end of the 5th week, mice were fasted to collect blood by orbital blooding, and then killed by cervical dislocation. Liver and white adipose tissues (inguinal white adipose tissue, iWAT; perirenal adipose tissue, pWAT; epididymal white adipose tissue, eWAT) were rapidly excised and weighed. Liver tissues were quickly frozen in liquid nitrogen and then stored at −80 °C for further analysis.

### 4.3. Fecal Microbiota Analysis

Total genome DNA from fecal samples were extracted using an E.Z.N.A.® Stool DNA Kit (Omega Bio-Tek Inc., Norcross, GA, USA), then DNA concentration and purity were monitored on 1% agarose gels. The V3-V4 region of the bacterial 16S ribosomal RNA gene was amplified using a specific primer (338F, 5′-ACTCCTACGGGAGGCAGCAG-3′; 806R, 5′-GGACTACHVGGGTWTCTAAT-3′). PCR was performed in a 20 μL mixture containing 4 μL 5×FastPfu Buffer, 2 μL of 2.5 mM dNTPs, 0.8 μL of Forward Primer, 0.8 μL of Reverse Primer, 0.4 μL of FastPfu Polymerase, 0.2 μL of BSA, and 10 ng of template DNA. The reactions were performed using a thermocycler PCR system and then amplicons were detected using 2% agarose gel electrophoresis and purified using an AxyPrep DNA gel extraction kit (Axygen Bioscience, San Diego, CA, USA) according to the manufacturer’s instructions. Quantified and purified amplicons were sequenced using the Illumina MiSeq platform (Illumina, San Diego, CA, USA) at Majorbio Bio-Pharm Technology Co., Ltd. (Shanghai, China) according to standard protocols. The raw reads were deposited into the NCBI Sequence Read Archive (SRA) database (Accession Number: PRJNA807051). The sequences were analyzed and assigned to operational taxonomic units (OTUs; 97% identity). Alpha diversity was analyzed using QIIME (Version 1.7.0), which included the calculation of Sobs, Shannon, Ace, Chao1, and Simpson indices.

### 4.4. SCFAs Detection

The fecal concentrations of acetate, propionate, butyrate, isobutyrate, and isovalerate were detected. Approximately 100 mg of feces was placed into a 1.5 mL centrifuge tube, dissolved with 1 mL ddH_2_O, and then centrifuged at 10,000 rpm for 10 min at 4 °C. Then, 0.9 mL of the supernatant was added to 0.1 mL 25% metaphosphoric acid solution, then vortexed for 1 min and centrifuged at 1000 rpm for 10 min at 4 °C after standing at 4 °C for over 2 h. The supernatant was then filtered using a 0.45 μm polysulfone filter and analyzed using an Agilent 6890 gas chromatograph (Agilent Tecnologies, Inc., Palo Alto, CA, USA).

### 4.5. Blood Glucose and Serum Parameter Analyses

Blood glucose and fasting blood glucose (after fasting for 6 h) concentrations were measured from blood obtained after tail vessel incision using an Accu-Check Performa glucometer (Roche). Serum samples were separated from the blood by centrifugation at 1000× *g* for 15 min at 4 °C and stored at −80 °C for further analysis. Serum TC, TG, LDL, HDL, and FFA levels were measured using an automatic biochemical analyzer (fully automatic bioanalyzer BK-280, Biobase Biodustry Co., Ltd, Shandong, China). Serum LPS, insulin, and leptin levels were measured using the ELISA assay (Nanjing Bioengineering Institute, Nanjing, China) following the manufacturer’s instructions.

### 4.6. Hepatic Histology Analysis

Liver tissues were fixed in 4% paraformaldehyde-PBS overnight, then were dehydrated and embedded in paraffin blocks. After that, a section of 5 μm was cut and mounted on slides. The sections were further deparaffinized, hydrated, and then stained with hematoxylin and eosin (H&E) for histological analysis. Images were captured using a DM300 microscope (Leica, Weztlar, Germany).

### 4.7. Hepatic Transcriptomics Analysis

Total RNAs were isolated using TRIzol® reagent (Invitrogen, Waltham, MA, USA), and contaminating genomic DNA was removed using DNAase I (Takara, Otsu, Japan). The RNA quality was determined with a 2100 Bioanalyzer (Agilent, alo Alto, CA, USA) and quantified with ND-2000 (NanoDrop Technologies, Wilmington, DE, USA). Oligo (dT)-enriched mRNA was fragmented by fragmentation buffer, and the cleaved RNA fragments were reverse transcribed to establish the final cDNA library using a SuperScript double-stranded cDNA synthesis kit with random hexamer primers (Illumina, San Diego, CA, USA). After being end-repaired, adenylated, and ligated to the sequencing adaptors, the RNA-seq sequencing library was sequenced on an Illumina Hiseq xten/NovaSeq 6000 sequencer (Illumina, San Diego, CA, USA) at Majorbio Bio-Pharm Technology Co. Ltd. (Shanghai, China). The data were analyzed using the online platform of Majorbio Cloud Platform (www.majorbio.com, accessed on 5 December 2021). Statistical significance of the differentially expressed data was defined with *p* adjust < 0.05 and fold change ≥ 2. 

### 4.8. Quantitative Real-Time PCR Analysis

Total RNA was extracted from liver samples using Trizol (Invitrogen, Carlsbad, CA, USA) reagent and then was treated with DNase I (Invitrogen, USA) according to the manufacturer’s protocols. The RNA samples (1 μg) were reverse transcribed to cDNA using PrimeScript Enzyme Mix 1, RT Primer Mix and 5 × PrimerScript Buffer 2 (Takara, Dalian, China). The reverse transcription was conducted at 37 °C for 15 min, 85 °C for 5 s. Gene-specific primer sequences (Table 2) were designed using Primer 5.0 software and synthesized by Sangon Biotech Co., Ltd. (Shanghai, China). Real-time PCR was performed according to the manufacturer’s instructions. Briefly, 1 μL cDNA template was added to a total volume of 10 μL containing 5 μL KAPA SYBR FAST qPCR Master Mix Universal, 0.4 μL PCR forward primer, 0.4 μL PCR reverse primer, 0.2 μL ROX low, and 3 μL PCR-grade water (Kapa biosystems, Beijing, China). The amplification procedure was as follows: 3 min at 95 °C; 3 s at 95 °C and 34 s at 60 °C for 40 cycles; 15 s at 95 °C, 1 min at 60 °C, and 15 s at 95 °C. Relative mRNA expression was normalized using reference gene β-actin and calculated using the 2^−ΔΔCt^ method.

### 4.9. Statistical Analysis

All statistical analysis was performed by using the one-way ANOVA with Duncan’s post hoc test (SPSS version 21.0, IBM Corp., Chicago, IL, USA). Spearman correlation analyses between the gut microbiota and serum metabolic parameters were conducted using GraphPad Prism 9.0. Data are expressed as the mean ± SEM. A *p*-value less than 0.05 was considered significant.

## 5. Conclusions

Collectively, the present study showed that gut microbiota depletion alleviated HFD-induced obesity and inhibited hepatic lipid accumulation, which might be associated with alterations in the expression of genes related to hepatic cholesterol and fatty acid metabolism. An Abx-induced microbiota depletion model is low cost and easier to establish relative to germ-free mice; however, to be precautious, the effects of Abx treatment on the host metabolism should be taken into consideration in the future. For our further research, there are three aspects that need to be considered. Firstly, a 5-week experimental period may be a little short for an HFD to exhibit remarkable effects on the gut microbiota and hepatic lipid metabolism. Secondly, some important microbiota-derived metabolites including SCFAs, BAs, and other metabolites in the serum or liver were not detected here, which should be further evaluated as mediators of gut microbiota in regulating liver metabolism. Thirdly, both depletion and supplementation (transplantation) of gut microbiota are necessary to evaluate the role of microbiota, which could provide a better and more definitive explanation.

## Figures and Tables

**Figure 1 ijms-23-09350-f001:**
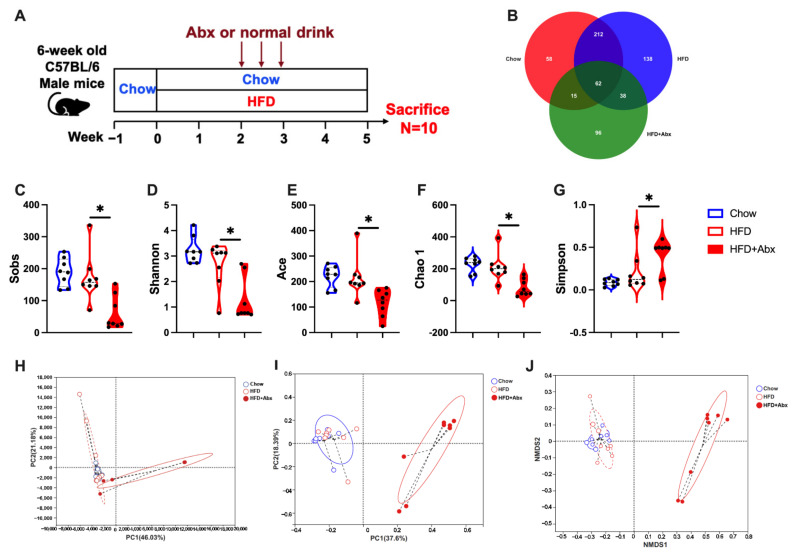
Abx treatment effectively depleted fecal microbiota. (**A**) Experimental design; (**B**) Venn diagram showing the unique and overlapping OTUs presented in Chow, HFD, and HFD+Abx groups; (**C**–**G**) α-diversity results; (**H**–**J**) β-diversity with PCA, PCoA, and NMDS score plots for discriminating the colonic microbiota from Chow, HFD, and HFD+Abx groups. * *p* < 0.05.

**Figure 2 ijms-23-09350-f002:**
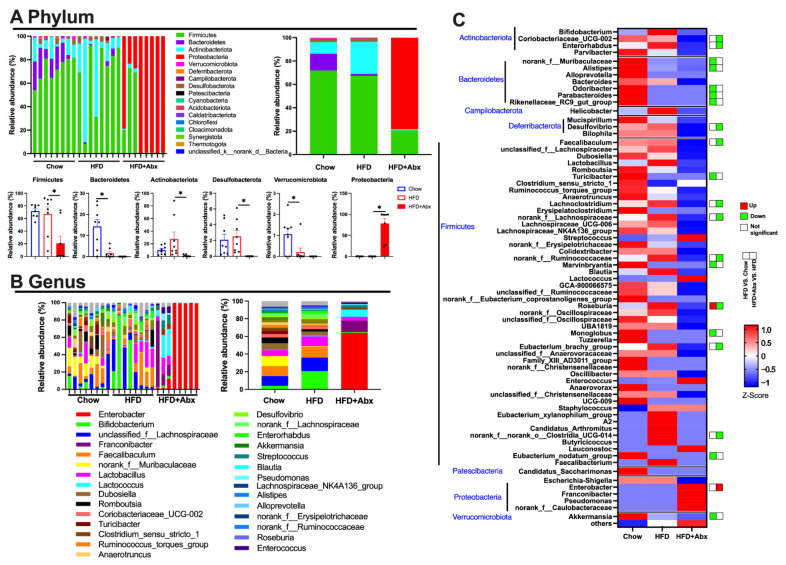
Gut microbiota depletion changed the microbial composition in HFD-fed mice. (**A**) Bacterial taxonomic profiling at the phylum level of fecal microbiota from different groups; (**B**) bacterial taxonomic profiling at the genus level of fecal microbiota from different groups; (**C**) heatmap of the bacterial taxonomic profiling at the genus level of fecal microbiota from different groups. Data were expressed as the mean ± SEM. * *p* < 0.05.

**Figure 3 ijms-23-09350-f003:**
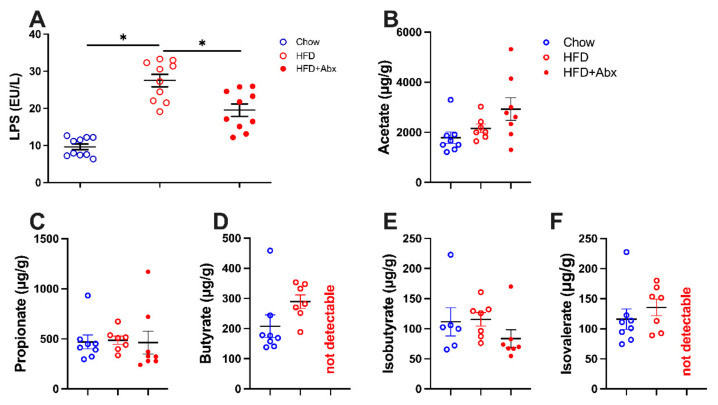
Gut microbiota depletion decreased the serum LPS and fecal SCFA content. (**A**) Serum LPS level; (**B**) fecal acetate content; (**C**) fecal propionate content; (**D**) fecal butyrate content; (**E**) fecal isobutyrate content; (**F**) fecal isovalerate content. Data were expressed as the mean ± SEM. * *p* < 0.05.

**Figure 4 ijms-23-09350-f004:**
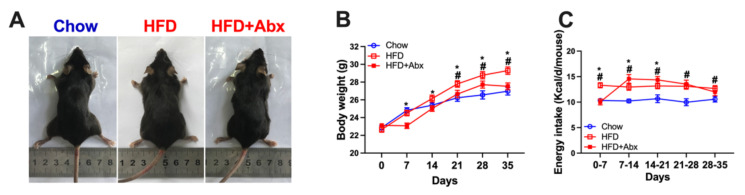
Gut microbiota depletion decreased body weight in HFD-fed mice. (**A**) Representative mice in the Chow, HFD, and HFD+Abx groups; (**B**) body weight; (**C**) energy intake. Data were expressed as the mean ± SEM. * *p* < 0.05 compared to HFD+Abx; ^#^
*p* < 0.05 compared to Chow.

**Figure 5 ijms-23-09350-f005:**
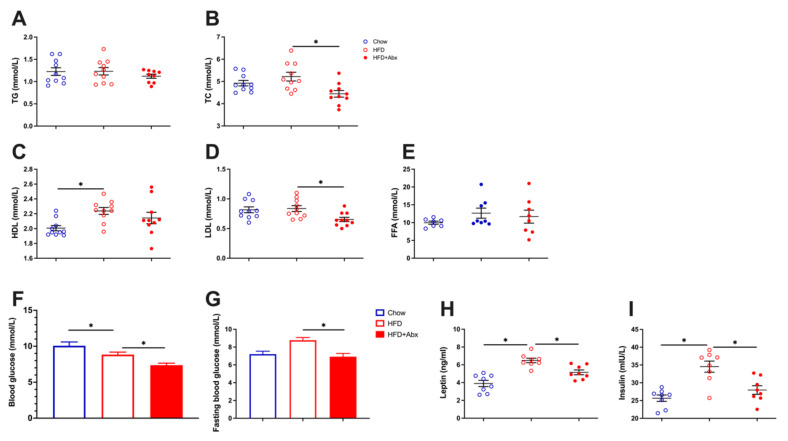
Gut microbiota depletion alleviated HFD-induced dyslipidemia. (**A**) Serum TG level; (**B**) serum TC level; (**C**) serum HDL level; (**D**) serum LDL level; (**E**) serum FFA level; (**F**) blood glucose concentration; (**G**) fasting blood glucose concentration; (**H**) serum leptin content; (**I**) serum insulin content. TC, total cholesterol; TG, triglycerides; LDL, low-density lipoprotein cholesterol; HDL, high-density lipoprotein cholesterol; FFA, free fatty acid. Data were expressed as the mean ± SEM. * *p* < 0.05.

**Figure 6 ijms-23-09350-f006:**
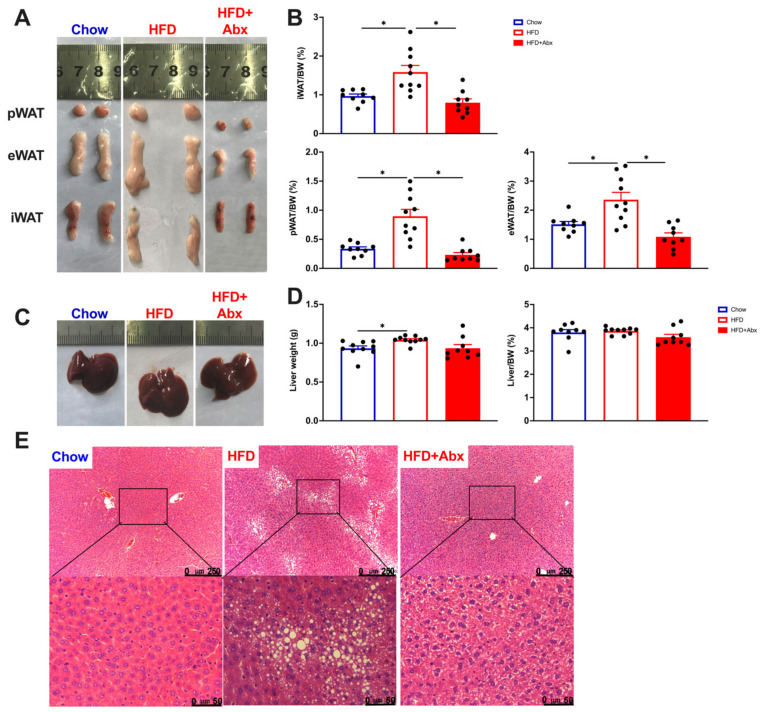
Gut microbiota depletion reduced fat deposition in HFD-fed mice. (**A**) Representative pictures for mice in the Chow, HFD, and HFD+Abx groups; (**B**) the ratios of iWAT, pWAT, and eWAT to body weight; (**C**) representative pictures for liver in mice from the Chow, HFD, and HFD+Abx groups; (**D**) liver weight and the ratio of liver to body weight; (**E**) representative pictures for H&E-stained liver sections. iWAT, inguinal white adipose tissue; pWAT, perirenal adipose tissue; eWAT, epididymal white adipose tissue. Small black squares are magnified in the bottom. Data were expressed as the mean ± SEM. * *p* < 0.05.

**Figure 7 ijms-23-09350-f007:**
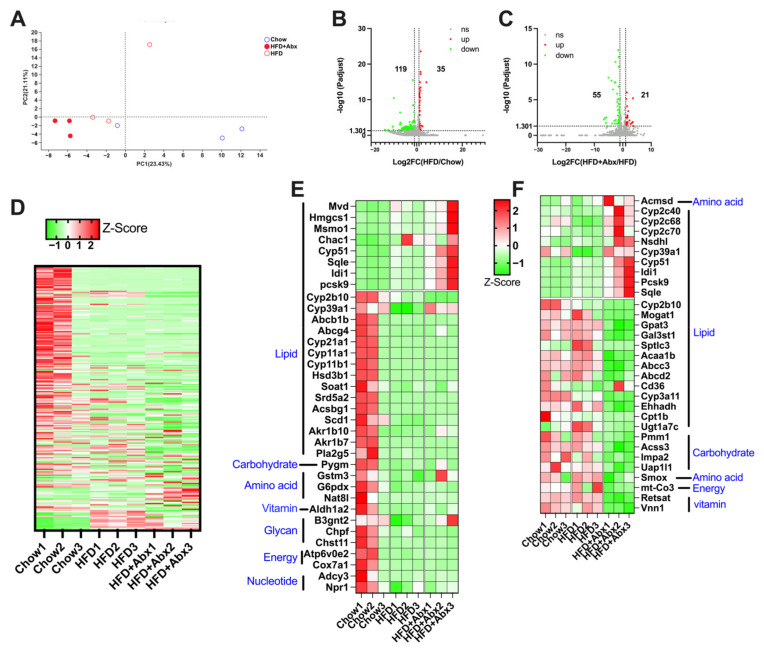
Gut microbiota depletion altered liver gene expression in HFD-fed mice. (**A**) PCA score plots for discriminating the liver gene expression from Chow, HFD, and HFD+Abx groups; (**B**) volcano plot of DEGs between mice fed a chow diet and HFD; (**C**) volcano plot of DEGs between mice with and without Abx treatment when fed an HFD; (**D**) heatmap of all the DEGs altered by HFD and Abx treatment; (**E**) heatmap of DEGs related to metabolism between mice fed a Chow diet and HFD; (**F**) heatmap of DEGs related to metabolism between mice with and without Abx treatment when fed an HFD. DEGs, differentially expressed genes.

**Figure 8 ijms-23-09350-f008:**
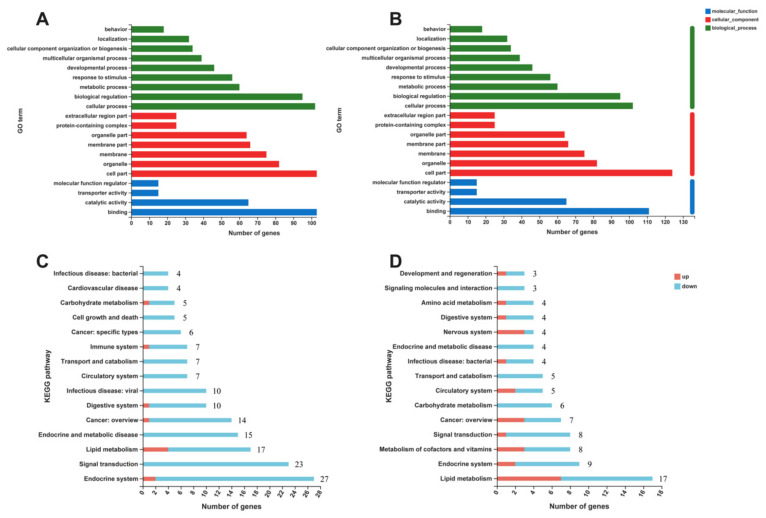
Gut microbiota depletion altered the expression of genes involved in lipid metabolism. (**A**) GO analysis of all differential genes that were altered by HFD feeding compared with chow diet fed mice; (**B**) GO analysis of all differential genes that were altered by Abx treatment in HFD-fed mice; (**C**) KEGG analysis of all differentially expressed genes that were altered by HFD feeding compared with chow diet fed mice; (**D**) KEGG analysis of all differential genes that were altered by Abx treatment in HFD-fed mice.

**Figure 9 ijms-23-09350-f009:**
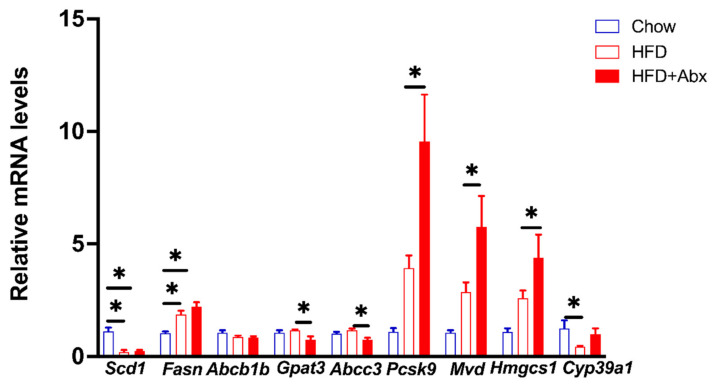
qRT-PCR validation of the differentially expressed genes between Chow, HFD, and HFD+Abx groups. Data were expressed as the mean ± SEM. * *p* < 0.05.

**Figure 10 ijms-23-09350-f010:**
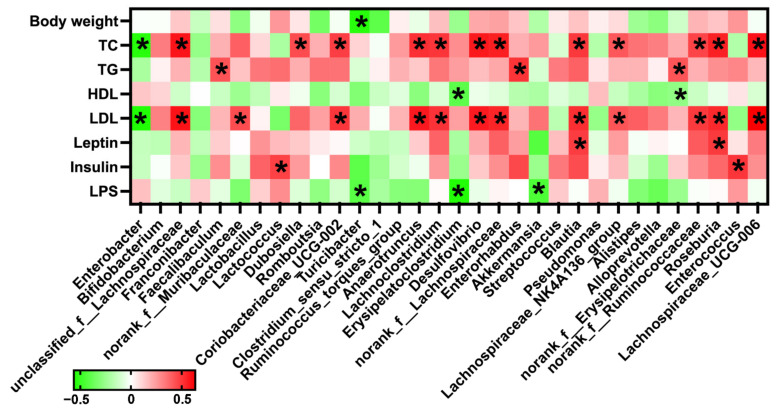
Heatmap result of Spearman correlation between the alterations in the fecal microbial population and the changes in serum lipid metabolic parameters. * *p* < 0.05 were significantly negative (green) or positive (red) Spearman correlation.

**Table 1 ijms-23-09350-t001:** Composition of normal chow diet and high-fat diet.

Ingredient (g/kg)	Chow Diet	High-Fat Diet
Casein	200	200
L-Cystine	3	3
Corn starch	351	0
Maltodextrin	35	125
Sucrose	350	68.8
Cellulose	50	50
Soybean oil	25	25
Lard	20	245
Minera mix	10	10
Dicalcium phosphate	13	13
Calcium carbonate	5.5	5.5
Potassium citrate, 1 H_2_O	16.5	16.5
Vitamin mix V10001	10	10
Choline bitartrate	2	2

**Table 2 ijms-23-09350-t002:** Primers used for qPCR assay.

Gene	Accession No.	Sequence (5′-3′)
β-actin	NM_007393.5	F: TGTCCACCTTCCAGCAGATGT R: GCTCAGTAACAGTCCGCCTAGAA
Scd1	NM_009127.4	F: CCTGCCTCTTCGGGATTTT R: GCCCATTCGTACACGTCATT
Fasn	NM_007988.3	F: CTCCACAGCTCTTCCAGTGAG R: ATGCTATTCTCTACCGCTGGG
Gpat3	NM_172715.3	F: GGGCAAGGGTAGAGTGTAGAAAA R: CGGAAAACACAGCGTCCAG
Abcb1b	NM_011075.2	F: AGTGGCTCTTGAAGCCGTAA R: AACTCCATCACCACCTCACG
Abcc3	NM_029600.4	F: ACTCTCATGTGGCGAAGCAT R: GATAAAGTCCGTCTGGGGCA
Pcsk9	NM_153565.2	F: ATCACCGACTTCAACAGCGT R: GCCCTTCCCTTGACAGTTGA
Mvd	NM_138656.2	F: CCACAACCGTTGCCATTAGC R: GTAGAGTGTCCCCGTCCTCT
Hmgcs1	NM_001291439.1	F: CTGATCCCCTTTGGCTCTTTCA R: CCCAGGCCGATGGTATACTT
Cyp39a1	NM_018887.4	F: TCACCAATAGCAATCGCCGT R: GAGGGGCTTTCCCAAACTCA

F, forward; R, reverse.

## Data Availability

Sequencing raw data on fecal microbiota of mice were deposited into the NCBI Sequence Read Archive (SRA) database (PRJNA807051, https://www.ncbi.nlm.nih.gov/bioproject/PRJNA807051). Sequencing raw data on hepatic transcriptomics were deposited into the NCBI Sequence Read Archive (SRA) database (PRJNA 832803, https://www.ncbi.nlm.nih.gov/bioproject/PRJNA832803).

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
