# Peer review of "Depletion of Gut Microbiota Inhibits Hepatic Lipid Accumulation in High-Fat Diet-Fed Mice"

_ijms, 2022, doi:10.3390/ijms23169350_

Round 1
Reviewer 1 Report
The authors performed experiments to determine the effects of antibiotic treatment to mice to kill the intestinal microbiota on the effects of a high fat diet, focusing on the effects on the liver. The experiments showed that the antibiotic treatment decreased weight gain and fatty liver. Global gene expression analysis of liver identified many changes in lipid metabolism genes that may be involved in the molecular mechanism. Overall, the experiments appear to be well-performed, and the paper is well-organized and well-written. However, there are a few minor comments.
Comment 1: As the authors write in the Conclusions, many of the effects of the antibiotic treatment may be independent of the changes in microbiota and just due to host toxicities from the combination of the 4 antibiotics. It would have strengthened the paper to have added a fourth group where the same dose of antibiotics was administered by i.v. or i.p. injection to mice fed the high-fat diet. This may largely block the killing of the intestinal bacteria, but maintain host toxicity. The weight graph and energy intake graph suggest that the antibiotics cause an initial toxicity that the mice then adapt to. Perhaps toxicity, RNA-Seq, or (Seahorse) respiratory experiments can be performed on isolated hepatocytes or HepG2 cells as a surrogate for in vivo effects on liver. Antibiotics frequently perturb mitochondrial protein synthesis that can lead to mitohormesis and potentially compensatory increases in fatty acid beta-oxidation potentially leading to a prevention of fatty liver such as the result observed. Although evidence for this was not observed at the gene expression level it could possibly occur through translational or post-translational mechanisms.
Minor comments (wording):
Abstract line 1: the key -> a key pathology
Abstract line 2: Gut -> The gut
Abstract line 2: confirmed to be correlated with -> shown to regulate
Abstract line 3: gut -> the gut
Abstract line 4: mice model -> mouse model
Abstract line 5: mechanism of -> mechanism through which
Abstract line 5: in regulating -> regulates
Introduction line 5: gut -> the gut
Introduction line 8: Gut -> The gut
Page 2 line 2: an high-fat -> a high-fat
Page 2 paragraph 2 line 6: immune -> the immune
Page 2 paragraph 2 line 11: transportation -> transport
Page 2 paragraph 3 line 1: As for the energy metabolism in host, gut -> The gut
Page 2 paragraph 3 second line from bottom: effects and mechanism of -> the effects of the
Page 2 paragraph 3 last line: a Abx-induced-> an Abx-induced
Page 2 paragraph 3 last line: was investigated -> were investigated
Figure 1 legend line 1: Veen -> Venn
Page 3 paragraph 1 line 12 -> Simpson -> the Simpson
Page 4 paragraph 1 line 9: Desulfobacterota, while increased -> and Desulfobacterota, while
it increased
Page 4 paragraph 1 second line from bottom: while increased -> while it increased
Figure 3 legend line 1: SCFAs contents -> SCFA content
Page 5 paragraph 1 line 1: were the microbial -> are microbial synthesized
Page 7 paragraph 2 line 1: Beside -> Besides
Figure 7 legend lines 2 and 3 (occurs twice): Volcan blot -> Volcano plot
Figure 8 legend lines 2 and 3 (occurs twice): Go -> GO
Figure 8 legend line 5: differential -> differentially expressed
Figure 9 legend line 1: validated the -> validation of the
Discussion line 8: have -> has
Discussion line 9: in host -> on the host
Discussion paragraph 2 line 6: effects -> effects on the host
Discussion paragraph 2 line 9: induced -> induces
Page 12 line 15: linolenic and -> linolenic acid
Page 12 line 16: acids -> acid
Page 12 line 29: blood circulating -> the circulation
Page 13 line 2: Besides -> In addition
Discussion 10 lines from bottom: bacterial -> bacteria
Discussion 5 lines from bottom: contains the major resistant bacteria -> is a major antibiotic-resistant genus
Discussion 4 lines from bottom: suggested -> suggest
Discussion second line from bottom: with the differential expressing analysis resulting from the -> differential expression analysis using
Discussion bottom line: RNA sequencing in the liver -> hepatic RNA sequencing
Section 4.2 title: experiment -> experiments
Page 14 line 9: was showed -> is shown
Page 14 line 13: weighted -> weighed
Section 4.4 line 2: tubes -> tube
Section 4.4 lines 3 and 5 (twice): under -> at
Section 4.4 line 7: Agilent 6890 gas chromatography -> an Agilent 6890 gas chromatograph
Section 4.5 line 4: under -> at
Section 4.7 line 2: DNA were -> DNA was
Section 4.7 line 3: 2100 Bioanalyser -> a 2100 Bioanalyzer
Conclusion line 4: to be established -> to establish
Conclusion line 7: 5 weeks’ -> 5-week
Conclusion fourth line from bottom: are necessary to -> should
Conclusion bottom line: confirmed -> more definitive
Reviewer 2 Report
The manuscript aims to identify the impact of gut microbiota and its depletion by antibiotics on regulating hepatic lipid metabolism in high-fat diet (HFD)-fed mice. The results of the animal study are well documented by charts and figures showing microbiota depletion-related changes in the bacterial composition and their impact on the expression of hepatic genes involved in metabolic pathways.
Overall, the manuscript is very well written. However, several issues might be addressed to improve the scientific quality of the article.
1. I would suggest placing the Figures with legends after the main text in each sub-section within the Results (2.1., 2.2., 2.3., 2.4., ...). That way it would be more appropriate and especially understandable for the readers. They will read the information first and then study the Figures.
2. I would suggest replacing mice fed a HFD with HFD-fed mice.
3. Description of chow diet within Material and Methods might be added.
4. In the Introduction - I recommend adding more information about the gut microbiome composition in health and disease. Moreover, it would beneficial to add a small paragraph concerning bacterial metabolites - SCFA (propionate, butyrate, acetate), their functions, and their impact on health.
5. Please, define the red rectangle bounding bacterial taxa Enterobacter in the Legend of Figure 2C.
6. Please, check if all abbreviations are explained in the main text, not only in the Figure legends (FFS, TC, TG, iWAT, pWAT, eWAT,...). It will be more clear for the readers if abbreviations will be also explained in the main text.
